# Performance Test of QU-Fitting

**Yoshimitsu Miyashita**

Graduate School of Science and Technology, Kumamoto University, 2-39-1, Kurokami, Kumamoto 860-8555, Japan; 170d9004@st.kumamoto-u.ac.jp

**Abstract:** QU-fitting is a model-fit method to reproduce the model of the Faraday Dispersion Function (FDF or Faraday spectrum), which is a probability distribution function of polarized intensity in Faraday depth space. In order to find the best-fit parameters of the model FDF, we adopt the Markov Chain Monte Carlo (MCMC) algorithm using Geweke's convergence diagnostics. Akaike and Bayesian Information Criteria (AIC and BIC, respectively) are used to select the best model from several FDF fitting models. In this paper, we investigate the performance of the standard QU-fitting algorithm quantitatively by simulating spectro-polarimetric observations of two Faraday complex sources located along the same Line Of Sight (LOS), varying the gap between two sources in Faraday depth space and their widths, systematically. We fix the frequency bandwidth in 700–1800 MHz and make mock polarized spectra with a high Signal-to-Noise ratio (S/N). We prepare four FDF models for the fitting by changing the number of model parameters and test the correctness of MCMC and AIC/BIC. We find that the combination of MCMC and AIC/BIC works well for parameter estimation and model selection in the cases where the sources have widths smaller than 1/4 Full Width at Half Maximum (FWHM) and a gap larger than one FWHM in Faraday depth space. We note that when two sources have a gap of five FWHM in Faraday depth space, MCMC tends to be trapped in a local maximum likelihood compared to other situations.

**Keywords:** magnetic fields; methods: data analysis; techniques: polarimetric

---

## 1. Introduction

This is a conference proceeding of "The Power of Faraday Tomography" based on Miyashita et al., 2019 [1].

Rotation Measure (RM) synthesis is an advanced technique that enables us to know magnetic fields along a Line Of Sight (LOS) direction using the Faraday Dispersion Function (FDF) (Burn 1966 [2]; Brentjens and de Bruyn 2005 [3]; Akahori et al., 2018 [4]). FDF represents the distribution of polarized intensity as a function of Faraday depth, which is proportional to an integration of thermal electron density and magnetic fields along the LOS. Compared with the conventional Faraday rotation technique, FDF gives us three-dimensional information about thermal/cosmic-ray electron densities, polarized sources, and magnetic fields when there is no magnetic field inversion along an LOS.

However, we have some technical problems for the reconstruction of FDF from observed polarized intensity because we can obtain only a limited frequency coverage of polarized emission in real observation. As a result, FDF becomes ill-determined, and we cannot distinguish between signal and noise when the observational noise is large. To improve the quality, several techniques have been proposed, for example: RM CLEAN, which removes the sidelobes of the dirty FDF (Hogbom 1974 [5]; Heald et al., 2009 [6]; Kumazaki et al., 2014 [7]; Miyashita et al., 2016 [8]), QU-fitting, which fits model Stokes Q and U parameters to observed Stokes Q, U parameters (O'Sullivan et al., 2012 [9]; Ideguchi et al., 2014,a [10]; Ozawa et al., 2015 [11]; Kaczmarek et al., 2017 [12]; Schnitzeler et al., 2018 [13]; Miyashita et al., 2019 [1]), and compressive sensing, which assumes a sparsity of FDF and optimizes

a solution using appropriate constraints (Li et al., 2011,a,b [14,15]; Andrecut et al., 2012 [16]). Sun et al. (2015) [17] reported a benchmark test of these techniques and that QU-fitting showed high scores in many situations, but the detailed test of QU-fitting has not been investigated yet.

　　QU-fitting needs to find a best-fit parameter set of a given fitting model. In a method of efficient parameter search, Markov Chain Monte Carlo (MCMC) has been widely used. However, it is known that the parameter search suffers from the "local maximum problem", where the best-fit parameter set is trapped in a local maximum of likelihood and cannot reach the global maximum of likelihood. Moreover, we prepare several models for the fitting and have to select the most suitable one. The Akaike Information Criterion (AIC) and Bayesian Information Criterion (BIC) are often used for the model selection, and they evaluate the balance between the fit to data and the simplicity of the model.

　　In this paper, we evaluate the capability of QU-fitting for parameter estimation and model selection through a series of simulations, which consist of preparing fitting models, making mock observation data, and searching for the best-fit parameter set in a given fitting model using MCMC and model selection with AIC and BIC. In Section 2, we explain the details of RM synthesis, QU-fitting, and the model setting. We show the results in Section 3, and we discuss the success/failure cases in Section 4. Finally, we give a summary in Section 5.

## 2. Model and Calculation

### 2.1. RM Synthesis

　　The observed complex linear polarized intensity $P(\lambda^2)$ is expressed as:

$$P(\lambda^2) = Q(\lambda^2) + iU(\lambda^2) = \int_0^\infty \epsilon(r)e^{2i\chi(r,\lambda^2)}dr, \tag{1}$$

where $Q$ and $U$ are the Stokes parameters, $\epsilon(r)$ is the synchrotron emissivity as a function of source position $r$, and $\chi$ is the observed polarization angle expressed as:

$$\chi(r,\lambda^2) = \chi_0(r) + \phi(r)\lambda^2. \tag{2}$$

　　The observed polarization angle is proportional to the squared wavelength $\lambda^2$ (Faraday rotation effect), and $\chi_0$ is the intrinsic polarization angle. Faraday depth $\phi$ is defined as:

$$\phi = 0.81 \int_r^0 n_e B_{||} dr' \quad \text{rad m}^{-2}, \tag{3}$$

where $n_e$ is the number density of thermal electrons in cm$^{-3}$, $B_{||}$ is the LOS component of the intervening magnetic field strength in $\mu$G, and $r$ is the physical distance to the target source in pc. We change the integration variable from $r$ to $\phi$ of Equation (1), then the complex polarized intensity $P(\lambda^2)$ is rewritten as the following formula:

$$P(\lambda^2) = Q(\lambda^2) + iU(\lambda^2) = \int_{-\infty}^\infty F(\phi)e^{2i\phi\lambda^2}d\phi, \tag{4}$$

where $F(\phi)$ is FDF, which represents the complex polarized intensity distribution in Faraday depth space (Brentjens and de Bruyn 2005 [3]). From this formula, $F(\phi) \equiv \epsilon(\phi)e^{2i\chi_0(\phi)}$ shows the intrinsic polarized source properties as a function of Faraday depth. Equation (4) is similar to the Fourier transform, so we generally obtain FDF by the inverse Fourier transformation as follows:

$$F(\phi) = \frac{1}{\pi} \int_{-\infty}^\infty P(\lambda^2)e^{-2i\phi\lambda^2}d\lambda^2. \tag{5}$$

However, the obtained FDF is incomplete because the observable wavelength coverage is limited. As a standard formalism, we multiply a window function $W(\lambda^2)$, for which $W(\lambda^2) = 1$ if $\lambda^2$ is in the observable bands and otherwise $W(\lambda^2) = 0$, as follows:

$$\tilde{F}(\phi) = \frac{1}{\pi} \int_{-\infty}^{\infty} W(\lambda^2) P(\lambda^2) e^{-2i\phi\lambda^2} d\lambda^2, \tag{6}$$

$$= \frac{1}{K} F(\phi) * R(\phi). \tag{7}$$

$\tilde{F}(\phi)$ is called dirty FDF since the limited wavelength coverage produces a sidelobe in intrinsic FDF. We can describe the dirty FDF using a convolution theorem between the intrinsic $F(\phi)$ and the Rotation Measure Spread Function (RMSF) $R(\phi)$, which is an inverse Fourier transformation of the window function:

$$R(\phi) = K \int_{-\infty}^{\infty} W(\lambda^2) e^{-2i\phi\lambda^2} d\lambda^2, \tag{8}$$

$$K^{-1} = \int_{-\infty}^{\infty} W(\lambda^2) d\lambda^2, \tag{9}$$

where $K$ is the normalization of RMSF. The Full Width at Half Maximum (FWHM) of $R(\phi)$ determines the accuracy of the reconstruction of the FDF and is expressed by:

$$FWHM = \frac{2\sqrt{3}}{\lambda_{max}^2 - \lambda_{min}^2}, \tag{10}$$

where $\lambda_{max}^2$ and $\lambda_{min}^2$ are the squared maximum and minimum wavelength, respectively.

## 2.2. QU-Fitting

QU-fitting is a model-fitting method to estimate FDF. We assume the model FDF $F_{mod}(\phi; \theta)$ given the model parameters $\theta$ and convert $F_{mod}(\phi; \theta)$ to model the polarized spectrum $P_{mod}(\lambda^2; \theta)$ using Equation (4) to compare with the observed spectrum $P(\lambda^2)$. The parameters $\theta$ are searched until $P_{mod}(\lambda^2; \theta)$ is evaluated as the best fit to $P(\lambda^2)$. In other words, the fitting tries to find $\theta$ that minimizes the chi-squared value,

$$\chi^2(\theta) = \sum_{i=1}^{N} \left( \frac{P_{obs}(\lambda_i^2) - P_{mod}(\lambda_i^2; \theta)}{\sigma_{noise}} \right)^2, \tag{11}$$

where $N$ is the number of frequency channels, $i$ is the individual channels, and $\sigma_{noise}$ is the noise on $P(\lambda^2)$. Note that the noise is assumed to be frequency-independent in this work for simplicity, while it can be frequency-dependent in real observations.

As the fitting method, we adopt the Metropolis algorithm of the Markov Chain Monte Carlo (MCMC), which is relatively easy to implement, but provides effective sampling of the parameters. The MCMC starts with generating candidate parameters $\theta'$ from the normal distribution with the previous parameters, $\theta_t$, as the average and the given step width as the variance. Then, the candidate is accepted with the probability $u$,

$$u = \min\left(1, \frac{L(\theta')}{L(\theta_t)}\right), \tag{12}$$

and the parameters are updated as $\theta_t + 1 = \theta_t$, or otherwise, the candidate is rejected and the parameters are unchanged as $\theta_t + 1 = \theta_t$. Here, $L(\theta)$ is the likelihood given the parameters and is expressed using $\chi^2$ as:

$$L(\theta) \propto \exp\left(-\frac{1}{2}\chi^2(\theta)\right). \tag{13}$$

In this work, we perform sampling with 20,000 samples after adjusting the step widths of each parameter to ensure the acceptance ratio of 30 percent. Since the convergence of parameters is one of the indicators that the parameters are in the local minimum, the 20,000 samples are updated every 1000 steps and assessed with a convergence check. To evaluate the convergence, we use Geweke's diagnostics (Geweke 1992 [18]), in which we regard the parameters as being converged if the following condition is satisfied,

$$Z = \frac{\bar{y}_A - \bar{y}_B}{\sqrt{V(y_A) + V(y_B)}} < z, \tag{14}$$

where $y$- and $V(y)$ are the average and variance of a parameter $y$ in the samples, respectively, and the subscripts $A$ and $B$ are the first 10 percent and last 50 percent sections of the samples, respectively. We adopt $z = 1.96$ as the Z-value, which corresponds to the significance level of five percent. When the samples of all parameters satisfy the convergence condition or the number of whole MCMC steps reaches the maximum regulation number, which is set to 100,000 in this work, the MCMC stops, and the 20,000 samples at the point are used for further analysis.

Another important procedure for the QU-fitting is the model selection. It often happens that we try fitting a single dataset with several models and then select the optimum model that provides the best goodness-of-fit to the data. In this work, we adopt two criteria, the Akaike Information Criterion (AIC) and Bayesian Information Criterion (BIC), which are expressed as:

$$\text{AIC} = -2 \log L(\theta) + 2k, \tag{15}$$
$$\text{BIC} = -2 \log L(\theta) + k \log n, \tag{16}$$

where $k$ is the number of parameters and $n$ is the number of data. The first term represents the minimum $\chi^2$ value that is obtained from the MCMC result, and the second term implies the penalty for the number of parameters, which discourages overfitting. The models with the lowest criteria values are selected as the optimum models.

## 2.3. Method and Models

Miyashita et al. (2016) [8] reported that RM CLEAN shows a great performance for point sources such as delta functions, but it does not work well for the diffuse sources. Therefore, we set a correct answer of the QU-fitting test simulation as following two Gaussian functions:

$$\begin{aligned} F(\phi) &= \frac{f_1}{\sqrt{2\pi\sigma_1^2}} \exp\left(-\frac{(\phi - \phi_1)^2}{2\sigma_1^2}\right) e^{2i\chi_{0,1}} \\ &+ \frac{f_2}{\sqrt{2\pi\sigma_2^2}} \exp\left(-\frac{(\phi - \phi_2)^2}{2\sigma_2^2}\right) e^{2i\chi_{0,2}}, \end{aligned} \tag{17}$$

where $f_1$ and $f_2$ are the absolute values of polarized intensity, $\phi_1$ and $\phi_2$ are the source positions in Faraday depth space, $\sigma_1$ and $\sigma_2$ are the widths of two sources in Faraday depth space, and $\chi_{0,1}$ and $\chi_{0,2}$ are intrinsic polarization angles of two sources, respectively. This "source" model corresponds to two independent diffuse polarized sources (Faraday complex sources) or two components within a source along a single LOS. We consider 10 source models for this test simulation varying the gap between two sources and the source widths systematically as follows:

- Gap: $\phi_2 - \phi_1 = 0.5$ (g1), 1.0 (g2), 2.0 (g3), 5.0 (g4), and 10.0 (g5) in units of FWHM of the RMSF ($\phi_1$ is fixed).
- Width: $\sigma_1 = \sigma_2 = 0.25$ (w1) and $\sigma_1 = 0.5$, and $\sigma_2 = 0.25$ (w2) in units of FWHM of the RMSF.

We fix other parameters like $f_1 = f_2 = 3$ (mJy), $\phi_1 = 0$ (rad/m$^2$), $\chi_{0,1} = 0$ (rad), and $\chi_{0,2} = \pi/4$ (rad). We label the source models, for example w2g3, in the case with $\phi_2 - \phi_1 = 2.0$ FWHM and

$\sigma_1 = 0.5$ and $\sigma_2 = 0.25$ FWHM. The two Gaussians of the g1 source model overlap because the gap is smaller than FWHM.

Then, we make 10 mock polarized spectra calculated from these 10 source models (Equation (4)). We select the frequency coverage from 700 MHz–1800 MHz, which is the full bandwidth of the Australian SKAPathfinder (ASKAP); this is because we compare the results of the early science program of the Polarization Sky Survey of the Universe's Magnetism (POSSUM, Gaensler et al., 2010 [19]) on ASKAP. The FWHM is 22.26 rad/m², calculated by Equation (10), and the maximum scale, which is the sensitive largest structure in Faraday depth space calculated by $\pi/\lambda_{min}^2$, is 113 rad/m², enough to be larger than FWHM. The channel width is 1 MHz, and the number of data is 1100. We add Gaussian noise with the mean of zero and the standard deviation of one to each channel of these mock polarized spectra.

We prepare four "fitting" models named G1, G2, G3, and G4, which consist of one to four Gaussian function(s), respectively. For simplicity, we test if AIC/BIC can select the correct number of parameters, and the G2 model should be selected ideally. In each fitting model (G1–G4), the best-fit parameters are estimated by MCMC, and AIC/BIC is calculated as described in the previous subsection. The initial values of MCMC are zero, and the parameter is searched in a limited parameter space ($\phi$: $-1000$–$1000$ (rad/m²), f: 0–200 (mJy), $\chi_0$: $-\pi/2$–$\pi/2$ (rad), $\sigma$: 0–200 (rad/m²)), where the periodic boundary condition is applied to the polarization angle. For each source model, we simulate QU-fitting 100 times with different random noise realizations. Thus, we perform QU-fitting, in total: (10 source models) × (100 realizations) × (4 fitting models) = 4000 times.

## 3. Results

We evaluate the results of the performance test by four criteria as follows.

- Criterion (i), convergence of MCMC: This criterion checks whether all of the model parameters with the G2 fitting model satisfy Geweke's diagnostics. Note that in the failure case, the MCMC reaches the maximum regulation number (100,000).
- Criterion (ii), calculated chi-squared values $\chi^2$: We check whether $\chi^2$ for the G2 fitting model is smaller than 2389 (within $3\sigma$ of the chi-squared distribution with $n = 2200$ degrees of freedom); in other words, this criterion checks whether the fitting model matches the mock data spectrum.
- Criterion (iii), model selection: This criterion checks whether AIC/BIC can select the G2 fitting model (correct fitting model).
- Criterion (iv), parameter estimation: We check whether all eight true parameters for the G2 fitting model ($\phi_i$, $f_i$, $\chi_{0,i}$, $\sigma_i$, i = 1, 2) are within $3\sigma$ of their calculated $L(\theta)$. We regard parameter estimation to be not successful if any one of the eight correct parameters are out of $3\sigma$ of their $L(\theta)$.

QU-fitting should be evaluated in the order of these criteria. Therefore, we define a categorization of the results to clarify at which step the QU-fitting fails to reproduce the source model, as follows.

- Class (A): All four criteria satisfy the success line.
- Class (B): Criteria (i), (ii), and (iii) are satisfied, but Criterion (iv) does not satisfy the success line.
- Class (C): Criteria (i) and (ii) are satisfied, but Criteria (iii) and (iv) do not satisfy the success line.
- Class (D): Criteria (i) is satisfied, but Criteria (ii), (iii), and (iv) do not satisfy the success line.
- Class (E): None of the four criteria satisfy the success line.

We consider the simulation results as the success cases of QU-fitting when the results are categorized into Class (A). We classify the rest cases as Class (Other); for example, Criterion (ii)–(iv) satisfy the success line, but Criterion (i) does not. Figure 1 shows the percentages in 100 realization results of the categorization in Classes (A)~(E) for the 10 source models.

Almost all of the simulations show Class (A) categorization (over 80%), that is QU-fitting is effective for the two Faraday complex sources along an LOS in the separation of two sources larger than 1 FWHM (g2–g5). AIC and BIC select the correct model in all of the source models, and we can find out that these information criteria could be reliable for the model selection in QU-fitting. We

find a super resolution in some cases (Section 4.1); two sources are resolved even if the separation is smaller than one FWHM. Comparing the success ratio of the w1g1 and w2g1 source models, w2g1, which has a different Gaussian width, has high scores more than two times that of w1g1. Meanwhile, QU-fitting fails in some specific situations displayed in yellow and red boxes; for example, the w2g4 source model fails in MCMC convergence with a percentage of 18%, although the separation is five FWHM (Section 4.2). Parameter estimation also suffers from local maximum trapping in the w1g1 and w2g1 source models at the rates 65% and 32%, respectively.

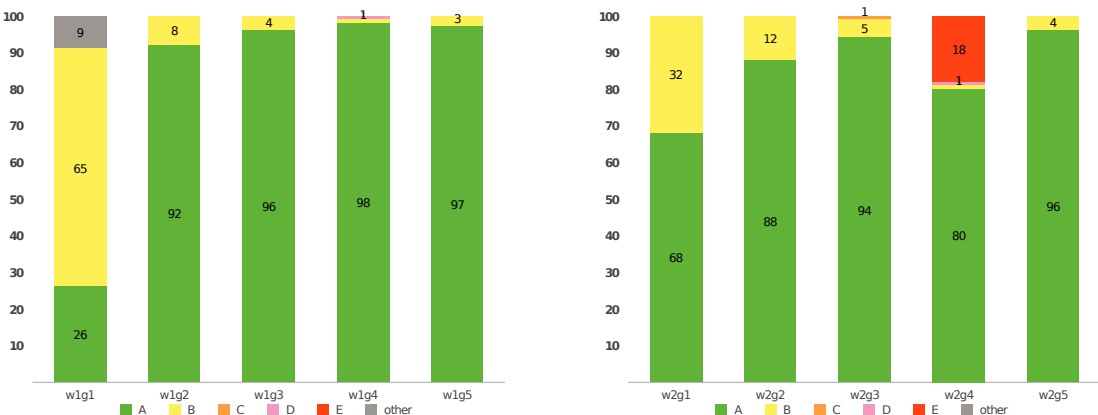

**Figure 1.** Categorization into Classes (A)~(E) of 100 simulation results for the Width 1 (w1) (left) and w2 (right) source models, respectively.

## 4. Discussion

In this section, we investigate the details of the unexpected success/failure cases of the QU-fitting simulation, and we fit w1g1 and w1g5 source models using the delta function model.

### 4.1. Success Cases within an FWHM Structure

The source models of w1g1 and w2g1 are two Gaussian functions whose gap in Faraday depth space is half the FWHM of RMSF. The widths of Gaussians were the same (w1g1, $\sigma_1 = \sigma_2 = 0.25$ FWHM) and different (w2g1, $\sigma_1 = 0.5$ FWHM, $\sigma_2 = 0.25$ FWHM). The maximum scale calculated from the inverse of the minimum lambda squared was 113 (rad/m$^2$) in the ASKAP band, so the widths were sufficiently small compared to it. Generally, it is difficult to resolve the FDF components within an FWHM resolution using a simple Fourier transform, but in the case of QU-fitting, AIC and BIC could separate the structures within an FWHM into two components with 26% (w1g1) and 68% (w2g1) probability. Figure 2 shows the comparison of the reconstructed FDF between RM CLEAN and QU-fitting in the w2g1 source model. The FDF of QU-fitting (G2 fitting model; selected by AIC and BIC) was one realization result categorized into Class (A). We can find that RM CLEAN could not resolve two sources, but QU-fitting was successful in the recovery of FDF with the two different widths. Although MCMC underestimated the peaks of FDF and overestimated the widths slightly, the correct model parameters were in the 3$\sigma$ range of calculated likelihood $L(\theta)$ (Equation (13)). QU-fitting showed good performance in the w2g1 compared to the w1g1 model, so the different amount of depolarization would be a key to identify these unresolved components.

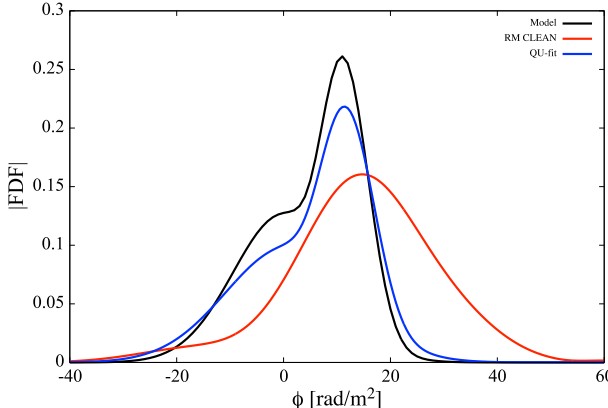

**Figure 2.** The reconstructed FDF by QU-fitting (blue line) of the w2g1 (g1, Gap 1) source model, cleaned FDF (red line) constructed by RM CLEAN using the same simulation data of QU-fitting, and the source model FDF (black line).

### 4.2. Failure Case of the Large Gap Model

As mentioned in the Results Section, MCMC could find the best-fit parameters correctly with high possibilities if the separation of two sources is larger than one FWHM. However, although the w2g4 source model satisfied this condition, this model showed bad performance with a percentage of about 20%. Figure 3 shows the distribution of the estimated Faraday depths of w2g4 in 100 realization simulations, and the color indicates the chi-squared values (Equation (11)) of the model-fit. Eighty percent of the simulations were successful, and the estimated Faraday depths were located at $\phi_1 = 0$ (rad/m$^2$) and $\phi_2 = 111.3$ (rad/m$^2$). The calculated chi-squared values were in the range of $3\sigma$ of the chi-squared distribution, and these values showed that the fitting model (G2 fitting model; the model selected by AIC/BIC) fit to the data well. On the other hand, 20% of the simulations failed, and the estimated Faraday depths were trapped in a local maximum area at $\phi_1 = \phi_2 = 70$–$100$ (rad/m$^2$). MCMC showed stochastic movement around the area, because the chain cannot be converged by the maximum step number. The calculated chi-squared values were out of $3\sigma$, so the fitting model G2 was quite different from the data. Figure 4 is the reconstructed FDF of RM CLEAN and QU-fitting, and the FDF of QU-fitting (G2 fitting model; not selected by AIC and BIC) was one realization result categorized into Class (E). Although RM CLEAN overestimated the source widths, it could estimate the location of the sources in the Faraday depth space due to the sufficiently large gap. However, the FDF of QU-fitting merged into a Faraday thick component, and AIC/BIC selected the wrong fitting model.

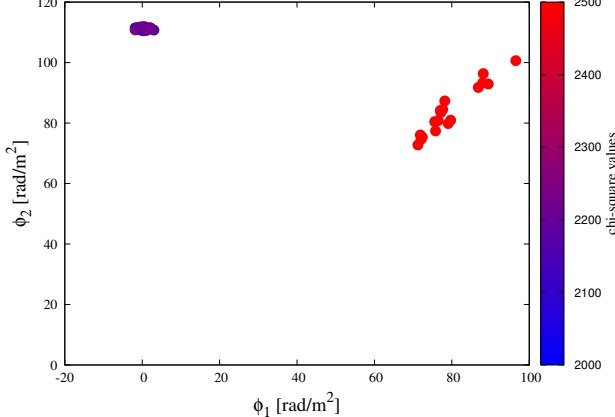

**Figure 3.** Correlation of the estimated best-fit Faraday depths of the w2g4 source model in 100 realizations. Color indicates the chi-square values calculated by Equation (11).

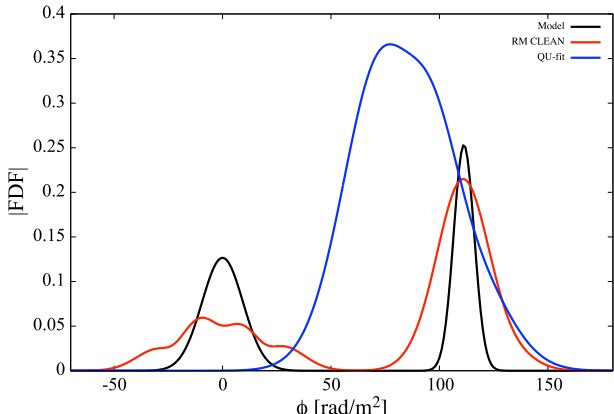

**Figure 4.** The reconstructed FDF by QU-fitting (blue line) of the w2g4 source model, cleaned FDF (red line) constructed by RM CLEAN using the same simulation data of QU-fitting, and the source model FDF (black line).

## 4.3. QU-Fitting Simulation Using the Delta Function Fitting Model

We verified that AIC/BIC can select the correct model even if we also did not have the prior information (not only the number of sources) about the shape of FDF. We added two fitting models, named d1 and d2, which consisted of one and two delta function(s), respectively. Physically, these fitting models had a different number of point sources along the LOS. Figure 5 is the reconstructed FDFs by the d1 and d2 fitting models for the w1g1 and w1g5 source models. In these fitting models, MCMC misestimated the peak position(s) and the total amplitude in Faraday depth space. Furthermore, for the w1g5 source model, these fitting models were reproduced in one Gaussian component ($\phi = 0$ (rad/m$^2$)) because the delta function(s) could not catch the oscillation produced by the larger Faraday depth component (see Figure 6). Therefore, AIC/BIC could select the correct model (G2 fitting model) in both source models because the calculated AIC/BIC values of the d1/d2 fitting models were larger than those of the G1~G4 fitting model (however, the calculated AIC/BIC values by the d2 fitting model in the w1g1 model were slightly larger than those of the G2 fitting model).

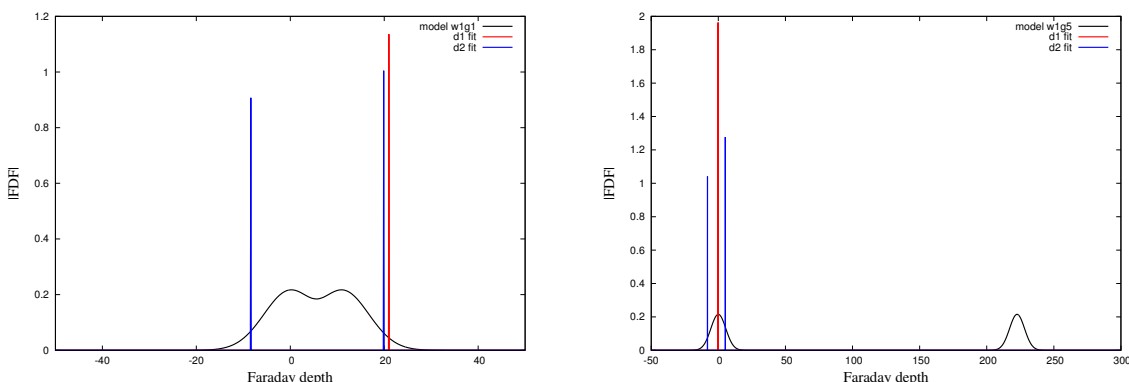

**Figure 5.** The reconstructed FDF by the Delta 1 (d1) (red) and d2 (blue) fitting models for the w1g1 (left) and w1g5 (right) source models, respectively. The black lines show the source model FDF.

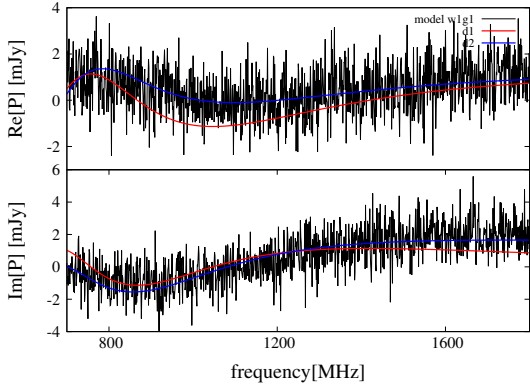 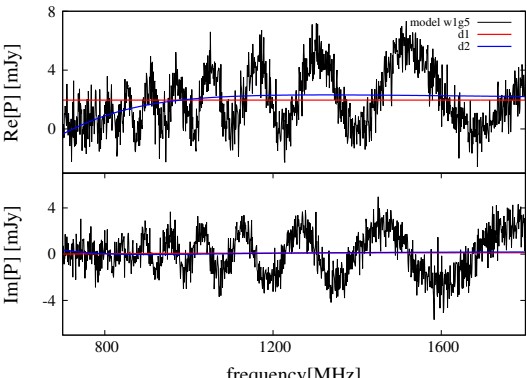

**Figure 6.** Estimated Stokes Q (top) and U (bottom) spectra by the d1 (red) and d2 (blue) fitting models for the w1g1 (left) and w1g5 (right) source models, respectively. These black lines show the source model Stokes Q,U spectra.

## 5. Conclusions

In this paper, we examined the functionality of the standard QU-fitting algorithm quantitatively by simulating spectro-polarimetric observations of two extent sources located along the same LOS. We assumed two Gaussian functions as a model of the extent sources and varied the gap and width systematically. We made simulation data supposing ASKAP observation (700–1800 MHz) by Fourier transformation of the FDF models. We prepared the following four criteria to evaluate the effectiveness of QU-fitting: (1) convergence of MCMC, (2) calculated chi-squared value, (3) model selection by AIC and BIC, and (4) parameter estimation by MCMC. For source models with a gap between two sources larger than one FWHM, QU-fitting worked well with high probability (>80%). Model selection was successful in almost all of the source models, and these criteria could select the correct model even if the source gap was smaller than one FWHM. However, parameter estimation sometimes suffers from the local maximum trapping problem, although the source gap was larger than one FWHM (w2g4 model).

**Funding:** This work is supported in part by a Grand-in-Aid from the Japan Society for the Promotion of Science (JSPS) Research Fellow, No. 17J06936 (YM).

**Acknowledgments:** We thank S. Ideguchi and K. Takahashi for useful discussions about this simulation.

**Conflicts of Interest:** The authors declare no conflict of interest.

## Abbreviations

The following abbreviations are used in this manuscript:

| | |
|---|---|
| RM | Rotation Measure |
| FDF | Faraday Dispersion Function |
| LOS | Line Of Sight |
| MCMC | Markov Chain Monte Carlo |
| AIC | Akaike Information Criterion |
| BIC | Bayesian Information Criterion |
| RMSF | Rotation Measure Spread Function |
| FWHM | Full Width at Half Maximum |
| POSSUM | Polarization Sky Survey of the Universe's Magnetism |
| ASKAP | Australian SKA Pathfinder |

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
