# Peer review of "Performance Test of QU-Fitting"

_galaxies, doi:10.3390/galaxies7030069_

Round 1

Reviewer 1 Report

The author presents a performance test of QU-fitting with MCMC method and model selection with AIC/BIC criteria. The test will be very useful for analyzing data from spectro-polarimetry. The results are well presented, and I suggest the paper be accepted for publication after minor revisions.

Major comments:

  The author fits the data with Gaussian which is the same as the input. It would be interesting to see the fitting results with alternative models such as delta functions.

Other comments:

1.) Abstract, the last sentence,

    I suggest it be reworded as,

    "We note that when two sources have ..."

2.) P1, Introduction, 2nd paragraph,

    "... FDF becomes dirty ..."

    change "dirty" to "ill-determined"?

3.) P2, Equation (2)

    If magnetic field is in unit of microG, distance is in unit of pc, and Faraday depth in unit of rad m^{-2}, the constant should be 0.81 instead of 811. Please check this.

    Please also add the limits for the integral.

3.) P2, Line 60,

    change such to similar.

4.) P3, Line 75,

    change that to which.

5.) P4, Section 2.3

    Please also describe the input values for f1 and f2.

6.) P4, Line 99,

    Please quote the formula for deriving the FWHM of 22.6.

7.) P4, Line 100,

    Please add a reference to POSSUM.

Author Response

p.p1 {margin: 0.0px 0.0px 5.0px 0.0px; line-height: 15.0px; font: 13.3px Arial; color: #000000; -webkit-text-stroke: #000000; background-color: #ffffff} p.p2 {margin: 0.0px 0.0px 5.0px 0.0px; line-height: 15.0px; font: 13.3px Arial; color: #000000; -webkit-text-stroke: #000000; background-color: #f4f4f4} p.p3 {margin: 0.0px 0.0px 5.0px 0.0px; line-height: 15.0px; font: 13.3px Arial; color: #000000; -webkit-text-stroke: #000000; background-color: #ffffff; min-height: 15.0px} p.p4 {margin: 0.0px 0.0px 5.0px 0.0px; line-height: 15.0px; font: 13.3px Arial; color: #000000; -webkit-text-stroke: #000000; background-color: #f4f4f4; min-height: 15.0px} span.s1 {font-kerning: none} span.s2 {font-kerning: none; background-color: #ffffff}

Other comments

1.) Abstract, the last sentence,

    I suggest it be reworded as,

    "We note that when two sources have ..."

Response : I changed the last sentence of abstract as follows, "We note that when two sources have ..." 

2.) P1, Introduction, 2nd paragraph,

    "... FDF becomes dirty ..."

    change "dirty" to "ill-determined"?

Response : I changed the word "dirty" to "ill-determined" in the text.

3.) P2, Equation (2)

    If magnetic field is in unit of microG, distance is in unit of pc, and Faraday depth in unit of rad m^{-2}, the constant should be 0.81 instead of 811. Please check this.

    Please also add the limits for the integral.

Response : I changed the constant to 0.81 and added the limits from x to 0 for the integral.

3-2.) P2, Line 60,

    change such to similar.

Response : I changed the word "such" to "similar" in the text.

4.) P3, Line 75,

    change that to which.

Response : I changed the word "that" to "which" in the text.

5.) P4, Section 2.3

    Please also describe the input values for f1 and f2.

Response : I added the explanation of these values in the line 89 as follows, "We perform simulations for 10 source models with fixed values of $f_1 = f_2 = 3$~[mJy], $\phi_1 = 0$~[rad/m$^2$], ...".

6.) P4, Line 99,

    Please quote the formula for deriving the FWHM of 22.6.

Response : I added the formula of FWHM in the end of section 2.1 and put that equation number in the line 98.

7.) P4, Line 100,

    Please add a reference to POSSUM.

Response : I added a reference (Gaensler et al. 2010) in the text.

Reviewer 2 Report

The work is very interesting and overall clear.
However, I had some thoughts while reading and maybe it would be good if the author could clarify them in the text.

You present at the very beginning of Sec 2.3 a model of two Gaussians, representing two polarized sources. However, later in this section (line 109) it is mentioned that you prepared 4 fitting models consisting of 1 to 4 Gaussians. So, maybe it would be better if you present one Gaussian function at the beginning and say that a multiple polarized component source is just the sum of N Gaussian, where N == 4. Therefore, G4= G1+G2+G3+G4.
Following this section it is also mentioned that the model G2 is the one that should be selected through AIC and BIC, could you please explain why?

The QU-fitting is very powerful and you show clearly that it is very good to identify two polarized component with a small separation; however, I don’t understand why it is not able to identify two components that are separated by 5 FWHM. Why only in this case the QU fitting is trapped in the local maximum area? Going from w2g1 to w2g3 there is an increase in the goodness of the fitting; then you have an abrupt drop when considering w2g4 and at the end, with w2g5, you encounter again an improvement in the goodness of the fitting. Why? Maybe add some words to clarify this..

What happened to the QU-fitting when considering only one Gaussian or more that two Gaussians?

The QU-fitting is also very powerful to identify complexity of the ambient medium, in the specific depolarization and whether the magnetic field is regular (with the presence of a gradient in the RM) or turbulent (with the presence os a Faraday dispersion sigmaRM). How these information can be extracted from your work? Is it possible to have not only the information about the number of polarized components, but also about the complexity of the magnetic field, e.g.? Maybe add few words on this issue.

Other comments:
_ Check the equation 2. The magnetic field B is in [mG], because of the constant 811.9. If using 0.81, then B is in [muG] unit
_ Line 101 “The channel width and the number are 1MHz and 1,100, …” clarify what is number. Number of data, I suppose.
_ Maybe try a synonymous for regard; it is used many times within the text.
_ Typo on line 71: “…the given step width as variance. we accept… “=> “…the given step width as variance. We accept…”
_ Typo in line 92 The separation of the two Gaussians. φ2 − φ1 = φ2 = 0.5 … maybe it was: φ2 − φ1 =  0.5

Author Response

p.p1 {margin: 0.0px 0.0px 0.0px 0.0px; line-height: 15.0px; font: 13.3px Arial; color: #000000; -webkit-text-stroke: #000000; background-color: #f4f4f4} p.p2 {margin: 0.0px 0.0px 5.0px 0.0px; line-height: 15.0px; font: 13.3px Arial; color: #000000; -webkit-text-stroke: #000000; min-height: 15.0px} p.p3 {margin: 0.0px 0.0px 5.0px 0.0px; line-height: 15.0px; font: 13.3px Arial; color: #000000; -webkit-text-stroke: #000000} p.p4 {margin: 0.0px 0.0px 0.0px 0.0px; line-height: 15.0px; font: 13.3px Arial; color: #000000; -webkit-text-stroke: #000000; background-color: #f4f4f4; min-height: 15.0px} p.p5 {margin: 0.0px 0.0px 0.0px 0.0px; line-height: 15.0px; font: 13.3px Arial; color: #000000; -webkit-text-stroke: #000000; background-color: #ffffff; min-height: 15.0px} p.p6 {margin: 0.0px 0.0px 0.0px 0.0px; line-height: 15.0px; font: 13.3px Arial; color: #000000; -webkit-text-stroke: #000000; background-color: #ffffff} p.p7 {margin: 0.0px 0.0px 5.0px 0.0px; line-height: 15.0px; font: 13.3px Arial; color: #000000; -webkit-text-stroke: #000000; background-color: #ffffff; min-height: 15.0px} p.p8 {margin: 0.0px 0.0px 5.0px 0.0px; line-height: 15.0px; font: 13.3px Arial; color: #000000; -webkit-text-stroke: #000000; background-color: #ffffff} span.s1 {font-kerning: none} span.s2 {font-kerning: none; background-color: #ffffff}

Comment : “You present at the very beginning of Sec 2.3 a model of two Gaussians, representing two polarized sources. However, later in this section (line 109) it is mentioned that you prepared 4 fitting models consisting of 1 to 4 Gaussians. So, maybe it would be better if you present one Gaussian function at the beginning and say that a multiple polarized component source is just the sum of N Gaussian, where N == 4. Therefore, G4= G1+G2+G3+G4.

Following this section it is also mentioned that the model G2 is the one that should be selected through AIC and BIC, could you please explain why?” 

Response : I was wondering you might misunderstand the simulation settings, so I’ll explain the detail and the purpose (I apologize for using “model” many times with different meanings). Firstly, I set FDF “source” model which consists of 2 Gaussians as the correct answer of this simulation test. Then, I prepared 4 models named as G1-G4 consisting of 1 to 4 Gaussian(s) for “fitting” models. MCMC estimates the best-fit model parameters for each given fitting model. Finally, I calculated AIC,BIC values for each fitting model and selected the lowest value of AIC,BIC as an optimum model among the 4 fitting models (G1~G4). This study checks whether AIC,BIC could select G2 model and QU-fitting could estimate the model parameters correctly. 

Comment : “The QU-fitting is very powerful and you show clearly that it is very good to identify two polarized component with a small separation; however, I don’t understand why it is not able to identify two components that are separated by 5 FWHM. Why only in this case the QU fitting is trapped in the local maximum area? Going from w2g1 to w2g3 there is an increase in the goodness of the fitting; then you have an abrupt drop when considering w2g4 and at the end, with w2g5, you encounter again an improvement in the goodness of the fitting. Why? Maybe add some words to clarify this.. “

Response : Firstly, I argue that QU-fitting could separate two components and estimate model parameters correctly with 80% probability in w2g4 source model. But I cannot specify the reasons why only this source model shows bad performance with 20% probability compared to other source models (randomness or signal to noise ratio, …). Actually, it is understood that the result is improved as changing the setting of the polarization angle even in the same situation (Miyashita et al. 2019).

Comment : What happened to the QU-fitting when considering only one Gaussian or more that two Gaussians?

Response :  I simulated QU-fitting for 1 Gaussian FDF model using the G1~G4 fitting models. Fig.1 shows the reconstructed FDF of G1 (red), G2 (blue), G3 (green) and G4 (purple) fitting models. The mean Faraday depth and amplitude of correct answer FDF (black) are 0 [rad/m^2] and 3 [mJy], respectively. AIC/BIC select the G1 fitting model (correct model) as the most optimum model. In the case of G2 and G3 fitting models, one component is reproduced correctly, while the other component(s) is located at large absolute Faraday depth with a large thickness.

Comment : The QU-fitting is also very powerful to identify complexity of the ambient medium, in the specific depolarization and whether the magnetic field is regular (with the presence of a gradient in the RM) or turbulent (with the presence os a Faraday dispersion sigmaRM). How these information can be extracted from your work? Is it possible to have not only the information about the number of polarized components, but also about the complexity of the magnetic field, e.g.? Maybe add few words on this issue.

Response : The purpose of this simulation is to evaluate the performance of QU-fit. Therefore, these problems are still working as future work. For example, there is a correlation between the FDF skewness and coherent magnetic fields (Ideguchi et al. 2017).

Other comments:

Comment :  Check the equation 2. The magnetic field B is in [mG], because of the constant 811.9. If using 0.81, then B is in [muG] unit.

Response : I changed the constant to 0.81.

Comment : Line 101 “The channel width and the number are 1MHz and 1,100, …” clarify what is number. Number of data, I suppose.

Response : I changed the text to “The channel width and the number of the data are 1MHz and 1100, respectively, …”.

Comment : Maybe try a synonymous for regard; it is used many times within the text.

Response : I changed (a part of) the word to “consider”.

Comment : Typo on line 71: “…the given step width as variance. we accept… “=> “…the given step width as variance. We accept…”

Response : I changed the “we” to capital character.

Comment : Typo in line 92 The separation of the two Gaussians. φ2 − φ1 = φ2 = 0.5 … maybe it was: φ2 − φ1 =  0.5

Response : I changed the equation to “φ2 - φ1 = 0.5”.

Round 2

Reviewer 2 Report

the manuscript 

Title: Performance test of QU-fitting in cosmic magnetism study

Authors: Yoshimitsu Miyashita

is fine and ready for publication.

Author Response

I apologize the delay in my revision.

I changed almost all of the sentences in this proceedings to avoid overlapping with the published paper.

I added the reference of the published 2019 MNRAS paper and I explained that this is a conference proceedings summarizing/based on the 2019 paper in the introduction.
